# Analysis of the Training Effect of a Nursing Undergraduate Course on the Management of Radiation-Related Health Concerns—A Single Group Experiment

**DOI:** 10.3390/ijerph17207649

**Published:** 2020-10-20

**Authors:** Hiromi Kawasaki, Satoko Yamasaki, Natsu Kohama, Susumu Fukita, Miwako Tsunematsu, Masayuki Kakehashi

**Affiliations:** 1Department of Public and School Health Nursing, Graduate School of Biomedical and Health Sciences, Hiroshima University, 1-2-3, Kasumi, Minami-ku, Hiroshima 734-8553, Japan; morisato@hiroshima-u.ac.jp (S.Y.); caijinxiaobin@gmail.com (N.K.); fukita1234@hiroshima-u.ac.jp (S.F.); 2Department of Health Informatics, Graduate School of Biomedical and Health Sciences, Hiroshima University, 1-2-3, Kasumi, Minami-ku, Hiroshima 734-8553, Japan; tsunematsu@hiroshima-u.ac.jp (M.T.); kakehashi@hiroshima-u.ac.jp (M.K.)

**Keywords:** disaster, nuclear disaster, undergraduate nursing student, evaluation of training, public health

## Abstract

After the 2011 Fukushima Daiichi accident, demands regarding nursing staff’s management of nuclear disasters have increased. This study aimed to evaluate a basic training program to teach undergraduate nursing students on how to deal with public anxiety following a nuclear disaster. We analyzed the data of 111 third-year nursing students attending a Japanese university. We set attainment goals in three domains (cognitive, affective, psychomotor) regarding their acquisition of consultation techniques for radiation-related health concerns. We compared pre- and post-class response scores on a knowledge test and calculated word frequencies in health consultation scenarios. The pre-class mean score was 12.2 out of 27 points and the post-class mean score (directly after the course) was 21.0, a significant increase (*p* < 0.001). The mean level of attainment in each of the three domains increased significantly from pre-test to post-test (*p* < 0.001). The variety and number of words in the health consultations also increased. Students attained the program goals for radiation-related health concerns in all domains. During disasters, there is a great need for nursing staff to administer physical and mental care to the public. This program was evaluated to be one wherein nursing staff can acquire fundamental knowledge about radiation in a short time.

## 1. Introduction

The Great East Japan Earthquake of 2011 resulted in enormous damage [1], and the accident it caused at the Fukushima Daiichi Nuclear Power Plant continued to impact people’s lives long afterward [2]. It has been reported that the greatest health impact of a nuclear accident is on matters of mental hygiene, including fear and anxiety about radiation [3], which may cause a wide range of health problems [4]. To allay the public’s anxiety about radiation, it is important to improve their ability to identify pertinent details from among vast quantities of information [5].

Medical professionals have to engage in effective communication to protect people’s health [6], by understanding how to communicate with members of the general public that have different levels of awareness about radiation [7]. In particular, communication is an important part of nursing care, which involves the promotion of psychological and social health [8]. In the context of nuclear disasters, helping the public effectively identify relevant radiation-related information is a part of the nursing role [9].

Regarding the accident at the Fukushima Daiichi Nuclear Power Plant, nursing staff could not sufficiently help local residents because they themselves did not have enough knowledge or information about radiation [10]. Further, they demonstrated a lack of communication skills [11]. Concerns about radiation subsequently spread among residents of other communities outside of the disaster site [12], and eventually, throughout the world [13]. This experience indicates the need to deal with concerns about health and radiation before a disaster occurs [14]. The public’s ability to identify relevant information can help prevent discrimination against exposed persons, and revitalize communities [15].

No matter when or where they work or their field of specialization, all nursing staff, including public health nurses and midwives, need to be armed with knowledge about radiation to be able to help people [16]. To prepare all nursing staff for nuclear disasters, the topic of radiation must be comprehensively tackled in undergraduate nursing science curricula [17], as opposed to the current scenario, where the focus has mainly been on protecting clinical nursing care patients from radiation [18]. In Japan, specialized additional learning at the working nurse level is provided for the nurse to become a radiation specialist. Ever since the accident at the Fukushima Daiichi Nuclear Power Plant, the demand for broad education about radiation as it relates to radiation therapy, protection from radiation and nuclear disasters in the field of nursing science has increased [19]. However, undergraduate nursing science education has thus far failed to adequately cover disasters involving nuclear energy [20]. This is thought to be because radiation-related accidents and disasters are infrequent, making the history of systematic education on emergency radiation treatments a short one [21].

It is important for nursing students to be knowledgeable about disasters and learn their role through disaster nursing education [22]. In practice, radiation education has been reported to result in nursing students learning about the importance of their professional role and understanding the necessity of knowledge about radiation [23]. Additionally, the Ministry of Education, Culture, Sports, Science, and Technology has indicated the need for a new curriculum in nursing education [24]. In the new curriculum, radiation nursing is included in disaster nursing practice, which is not limited to radiation disasters. However, not enough time was allocated to learning as compared to the content of the curriculum. Radiation-focused learning is not effective for pre-licensed nursing students because students do not consider the basics of physical radiation to be directly related to nursing care. There is no clearly delineated method to teach nursing staff on how to help increase this knowledge in the general public. As members of the public and also as future nursing staff, nursing students need to acquire fundamental knowledge about radiation and recognize pertinent disaster information and help the public in this regard [25]. After obtaining a nurse license, they need a foundation for additional learning about radiation expertise and nursing care.

The purpose of this study is to develop and evaluate methods for learning disaster-related introductory knowledge and care regarding radiation for nursing students attending Japanese universities. The focus of the method is to learn about radiation through nursing care, which is of great interest to nursing students.

## 2. Materials and Methods

### 2.1. Design

This study employed a single-group pre-post quasi-experimental design.

### 2.2. Recruitment

For this study, conducted at a university in Hiroshima, we recruited nursing undergraduates from a class that had 65 and 60 third-year students enrolled in 2017 and 2018, respectively. Both groups took the same class in July of their third year. This study used secondary data (the lesson assignment data). During the third year, the month of July was three months since nursing undergraduates started studying specialized nursing care related to clinical nursing. We informed the students of the objective of the study, the procedures involved, their freedom to participate or decline participation, and how their data would be handled. The data of students who provided written informed consent were tagged with ID numbers to ensure anonymity and used for analysis. This study was conducted with the approval of the university’s ethics committee (Approval No. E-809-1).

### 2.3. Overview of Training

#### 2.3.1. Learning Objectives

Based on Bloom et al.’s taxonomy of educational goals [26], we defined attainment goals in three domains with regard to the acquisition of techniques for guidance on radiation-related health concerns. These goals were created by a group of 60 public health and school nurses with reference to basic radiation training goals for nursing professionals [27]. The attainment goals are as follows.

Cognitive domain: can understand the means of collecting information about radiation; can understand basic information about radiation.

Affective domain: can imagine the health concerns people might have; is interested in the information and techniques necessary to respond to them.

Psychomotor domain: can think of actions to prepare for a nuclear disaster; can think of ways to address people’s health concerns.

#### 2.3.2. Learning Content

The course involved consultations with guardians who were worried about their children’s internal exposure. In an example session, which was video-recorded live, a local resident came to a nursing professional (student) for advice. It had been three months since the accident, and they were over 100 km away from the power plant. To the nursing student, the guardian said, “I am worried that my son will inhale a cloud of dust playing outdoor sports on the school grounds. Do you have any countermeasures? For example, can you cancel extracurricular activities? My child hates voluntarily taking sick days from sports”.

#### 2.3.3. Teaching Flow

In a 90 min class, 30 min were devoted to the lecture and 30 to role play. The whole process is depicted in Figure 1.

#### 2.3.4. Lecture

The key contents of the lecture are radioactive substances (iodine-131 and cesium-137) and units of radiation (Becquerel (Bq) and Sievert (Sv)), natural and artificial radiation, internal and external exposure, half-life, metabolism, stable iodine tablet, hot spot, dosimetry, and the methods of relevant information collection.

### 2.4. Tools

#### 2.4.1. Knowledge Test

Based on the learning objectives determined by the faculty, a 27-item test comprising 20 true-false items and seven short-answer items was developed (Appendix A). A few example items include “Choose a location that is likely to be a radiation hotspot from the examples”, “Choose behaviors that may lead to internal exposure”, and “Write down the half-life of iodine-131”. The same test was administered before and after the class. Each question was worth 1 point, and the results were tallied to calculate a final score. We used Wilcoxon signed rank test to compare pre- and post-class responses.

#### 2.4.2. Self-Assessment with Regard to Attainment Goals

For self-assessment, we used 12 items rated on a five-point Likert scale, with the following response options—1: Not at all true; 2: Slightly true; 3: Moderately true; 4: Fairly true; and 5: Very true. Scores were expressed positively such that the higher the score, the greater the level of attainment. Goal achievement level was evaluated according to the cognitive, affective, and psychomotor domains. We compared the pre- and post-class response scores for each item using a Wilcoxon signed rank test.

#### 2.4.3. Worksheets

Students were given worksheets to write about how they would respond in the example scenario. Worksheets #1 and #2 were identical; the former was handed out before the class and the latter afterward. The changes in the learners’ responses to the example were used to evaluate the effect of the lesson, particularly in the psychomotor domain. We used text mining to analyze the free-response portion. Based on the view that the number of nouns represents the number of concepts [28], we used the nouns in each worksheet as the target for our analysis [29]. Multiple nouns with similar meanings were considered synonyms or near-synonyms and consolidated under a single noun. For example, mother, parent, caregiver, and guardian were all counted under guardian.

### 2.5. Data Analysis

For the analysis, we included those questionnaires that did not contain missing responses. We compared the responses of the 65 students who took the class in 2017 and the 60 who took it in 2018 and consolidated the groups into one after confirming the absence of any significant intergroup difference. Finally, we analyzed 110 students’ knowledge tests and self-assessments (88.0%) and 111 students’ worksheets (88.8%). The knowledge and self-assessment tests were analyzed with SPSS version 25.0 (IBM Corp, Armonk, NY, USA), using the Wilcoxon signed rank test, with the level of significance set at 0.05. For the analysis of the worksheets, we used IBM SPSS Text Analytics for Surveys 4.0 (IBM Corp, Armonk, NY, USA).

## 3. Results

### 3.1. Cognitive Domain Evaluation

The combined pre-class mean score was 12.2 points and the post-class mean score was 21.0, demonstrating a significant increase using Wilcoxon signed rank test (*p* < 0.001). We used three statements to survey the students’ level of attainment of these goals. The maximum score for each statement was 5, for a total of 15. The combined pre-class mean total score was 6.0 points and the post-class mean score was 10.4. The post-test scores were significantly higher than the pre-test (*p* < 0.001) (Table 1).

### 3.2. Affective Domain Evaluation

We used three statements to identify the students’ level of attainment of these goals. These are presented in Table 1. The combined pre-class mean total score was 7.0 points, and the post-class mean score was 9.1.

### 3.3. Psychomotor Domain Evaluation

We used six statements to determine the students’ level of attainment of these goals. These are presented in Table 1. The maximum score for each statement was 5, for a total of 30. The combined pre-class mean total score was 11.2 points, and the post-class mean score was 14.6.

In addition, text mining was also used to assess the psychomotor domain. We counted the nouns written on the worksheets through text mining; these are shown in descending order of frequency in Figure 2 and Figure 3. The number associated with a given noun is the number of unique respondents who used it on their worksheets, which helped avoid overlaps [29]. Before the lecture, the nouns used in the example scenario were quoted often, with the most frequent one being worry. This was also true after the lecture, although the frequency was much higher. The noun worry was used as an expression of sympathy with the example consultee. The nouns that only entered the top 20 after the lecture included metabolism and emission. The frequencies of school, confirmation, teacher, and consultation, all of which were used by the students to recommend that the consultee seek someone else’s advice, decreased after the lecture. Meanwhile, metabolism and emission were used to explain to the consultee how the body metabolizes radioactive substances inside it.

## 4. Discussion

### 4.1. Cognitive Domain

In this domain, the focus of the assessment was whether the student had acquired the basic radiation knowledge needed to reduce consultees’ health anxiety. The selected key contents are radioactive substances (iodine-131 and cesium-137) and radiation units (Bq, Sv), internal exposure, half-life, metabolism, hot spots, dosimetry, and the methods in the collection of relevant information. Iodine thyroid blocking using stable iodine tablets was added to the key contents. For student motivation, it is important to strictly select the knowledge necessary for case consultation. Further, choosing the contents needed for a case by themselves from a repository of knowledge is the next level for students.

The post-class scores were significantly higher, indicating increased knowledge. The scores on the statements relating to cognitive attainment goals were also significantly increased. If we consider just the example scenario, 30 min were sufficient for students to acquire the necessary knowledge. Nursing staff who support people in various situations need the intellectual training to become familiar with the academic knowledge base and acquire the capacity to think in ways important to the profession [30]. Registered nurses (RNs) have ranked learning anatomy and biology as more important than other subjects [31]. To become an RN, one needs a bachelor’s or associate degree, both of which require preparatory classes in math, chemistry, and biology [32]. However, knowledge about radiation is part of physics [33], a subject whose importance the RNs are not aware of. The low scores on the pre-class radiation knowledge test are likely related to the idea that there is little need for such knowledge in nursing practice. The example scenario required students to think about how they would respond to someone who was worried about radiation because of a lack of sufficient knowledge about it. The students assessed the scenario and provided information to the consultee; knowledge about radiation was thus an indispensable part of the support. The nursing students were deemed to have understood methods of gathering information about radiation and the necessity of that information as nursing staff, which was made possible by the learning activity in this study. 

Guiding people who lack sufficient knowledge tests RNs’ own knowledge, so continuous learning is important for them to be able to provide better care [31,34]. The analysis of the worksheets showed that students used more words in worksheet #2. The change in the number of nouns in particular and words in general reflected a change in cognition. There was a high correlation between the number of nouns and the number of concepts; as such, the number of nouns is important in text mining [28]. Thus, students acquired the necessary knowledge and attained the cognitive domain goals.

### 4.2. Affective Domain

In this domain, the students rated their level of interest in certain activities that were reflective of their ability to reduce consultees’ health anxiety. Thus, the students’ interest in others’ health concerns about radiation was assessed in the affective domain. The scores for “I can explain what aspects of health people find concerning” and “I have tried to come to my own conclusion about the meaning of the measured values that have been published” increased significantly. Students learned the necessity of understanding consultees’ concerns by thinking about how to help in the example scenario, and they recognized the importance of understanding the information they gathered from the perspective of a nursing professional. In addition, they thought about how to interpret the information they gathered and use it to provide appropriate support, although no significant difference was observed with regard to the statement “I am interested in news and reports about radiation”. In the class, the example scenario was concerned with a mother’s concerns about radiation, which was based on the observed increase in consultations for health problems associated with the rising concern for children’s health after the 2011 nuclear accident [35]. Research has shown that it is important for nursing staff to give informed advice to people in order for them to make decisions. [11,36,37]. The example scenario is a case where health concerns were brought about by radiation, and it reflects a setting where nursing professionals were actually made aware of their insufficient skills in a nuclear accident. The example allowed the nursing students to experience this firsthand, which is what we imagine led to the learning effect [16].

Students’ interest in situations involving radiation on the news did not change. A case study with multiple episodes increased learners’ interest and curiosity significantly more than just a single episode [38]. Finding ways to add information about people’s lives such as the example and to increase students’ interest in those people’s living environments may also get them interested in daily news.

### 4.3. Psychomotor Domain

Here, we assessed whether students had the ability to engage in activities to reduce consultees’ health anxiety. The scores on all items relating to the psychomotor attainment goals increased significantly. Though RNs are aware of their role in disasters, there has been no progress in the education to prepare them for possible situations [39,40]. That being said, students were able to equip themselves with the behavioral awareness represented by the attainment goal “will take action to prepare for a nuclear disaster” through role play. We also assumed that students’ scores on “can respond to people’s health concerns” improved by repeatedly thinking about how to respond in the example scenario.

The number of characters and nouns students used in their worksheets for how they would respond to the consultee increased substantially after the class. The acquisition of knowledge about radiation helped make their responses more specific and detailed. Before the class, nouns like school, confirmation, teacher, and consultation, which defer the response to a third party, were frequently used; this decreased after the class. The number of people using nouns related to radiation increased and the nouns metabolism and **emission** newly appeared. This indicates that they gained some knowledge of the behavior of radioactive substances following internal exposure by understanding their biological half-life. Meanwhile, nouns like worry and fuan (unease/worry/anxiety) represent the emotional dimension [41]. The dramatic increase in the number of students using such words after the class suggests that more students sympathized with the consultee in the example scenario. Students were deemed to have achieved the attainment goals upon the qualitative evaluation of their response to the consultee.

## 5. Conclusions

In this study, an undergraduate class on radiation-related health concerns was observed to have an effect on students’ basic knowledge about radiation and their response in a consultation scenario. We found that the students’ recognition of radiation’s high degree of relevance to nursing care was connected to their motivation to learn its basics. Thus, the learning objectives were attained. 

However, the results must be interpreted in the context of certain limitations. First, there is a need for longitudinal investigations to confirm the long-term efficacy of the class as well as the generalizability of the results to all nursing students. Second, the target group must be expanded in future studies, as students’ knowledge about and interest in radiation may be influenced by environmental differences, such as personal experience as a disaster victim and whether there is a nuclear power plant where their university is located or where they grew up. This study was conducted in a one-group pre-test and post-test design, without a control group, which leverages the usual lesson outcomes. This limitation should be taken into account when interpreting the results of this study. Despite these limitations, this short class could be considered as a potential part of a public health nursing exercise in undergraduate education.

Health education is an effective opportunity for nursing students to better understand radiation. It is necessary to consider whether there are other opportunities to learn about radiation in the workplace.

## Figures and Tables

**Figure 1 ijerph-17-07649-f001:**
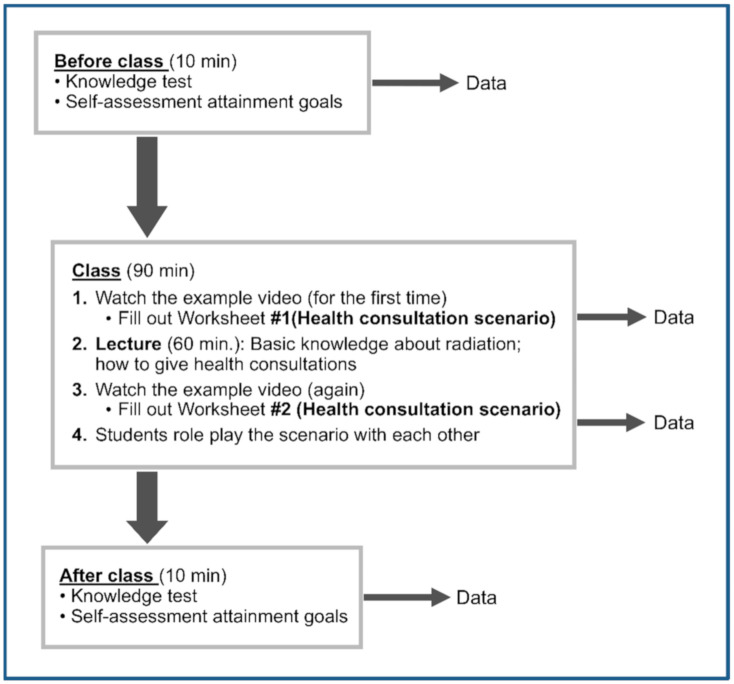
Flow and teaching tools used in the class.

**Figure 2 ijerph-17-07649-f002:**
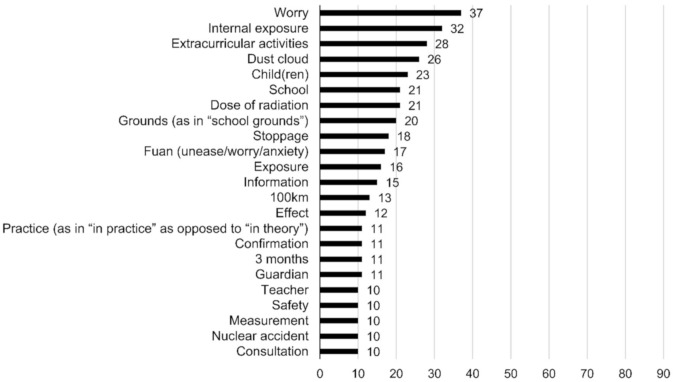
The nouns that students wrote on the pre-class worksheet with regard to how they would respond to the example scenario and the number of students who wrote them.

**Figure 3 ijerph-17-07649-f003:**
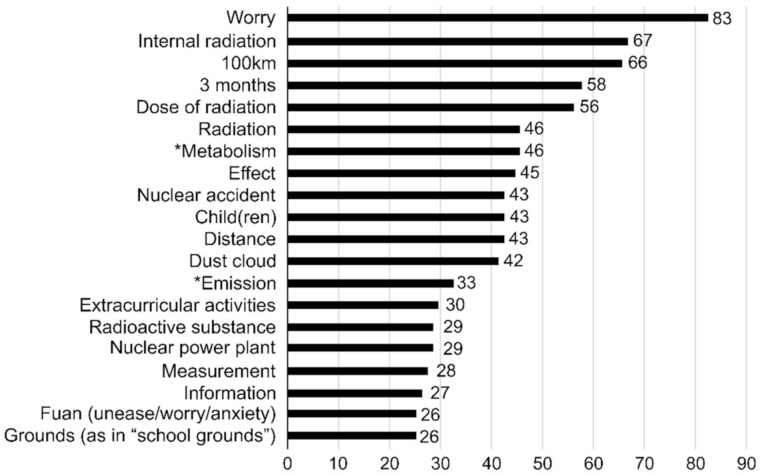
The nouns that students wrote on the post-class worksheet with regard to how they would respond to the example scenario and the number of students who wrote them. Note: words that did not appear in worksheet #1 are marked with an asterisk (*).

**Table 1 ijerph-17-07649-t001:** Level of attainment on three domain goals before and after the class.

Learning Domain	Item	*n*	Before		After		*Z*	*p*-Value
			Mean	*SD*	Mean	*SD*		
Cognitive	I know where I can obtain information about radiation.	109	1.9	0.9	3.0	1.1	−7.711	*p* < 0.001
	I know the government has released the results of studies on radiation.	110	2.5	1.2	3.6	1.2	−6.622	*p* < 0.001
	I can explain what natural radiation is.	110	1.6	0.8	3.9	0.9	−8.888	*p* < 0.001
	Total level of attainment (maximum score: 15)	109	6.0	2.1	10.5	2.5	−8.917	*p* < 0.001
Affective	I can explain what aspects of health people find concerning.	110	2.4	0.8	3.3	1.0	−6.818	*p* < 0.001
	I am interested in news and reports about radiation.	109	2.7	0.9	2.8	0.9	−1.381	*p* = 0.167
	I have tried to come to my own conclusion about the meaning of the measured values that have been published.	110	2.0	0.9	3.0	1.0	−6.710	*p* < 0.001
	Total level of attainment (maximum score: 15)	109	7.0	2.0	9.1	2.4	−7.091	*p* < 0.001
Psychomotor	I have thought about methods of obtaining accurate information to prepare for a nuclear disaster.	110	2.2	0.9	2.7	1.0	−4.815	*p* < 0.001
	I have thought about protecting myself against a nuclear disaster.	110	2.2	1.0	2.5	0.9	−3.074	*p* = 0.002
	I have thought about how to respond during a nuclear disaster.	110	2.4	0.9	2.7	1.0	−2.727	*p* = 0.006
	I have thought about the items necessary at workplaces during a nuclear disaster.	108	1.7	0.8	2.1	1.0	−4.559	*p* < 0.001
	I do nuclear disaster training elsewhere outside of class.	110	1.2	0.6	1.5	0.8	−3.595	*p* < 0.001
	I can use my knowledge about radiation to respond to the health concerns that people bring up.	110	1.6	0.7	3.2	0.9	−8.602	*p* < 0.001
	Total level of attainment (maximum score: 30)	108	11.2	3.4	14.6	4.1	−7.841	*p* < 0.001

Wilcoxon signed-rank test.

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
