# Peer review of "Analysis of the Training Effect of a Nursing Undergraduate Course on the Management of Radiation-Related Health Concerns—A Single Group Experiment"

_ijerph, 2020, doi:10.3390/ijerph17207649_

Round 1
Reviewer 1 Report
Kawasaki et al. evaluated a basic training program to teach undergraduate nursing students how to deal with public anxiety following a nuclear disaster. They compared pre- and post-class response scores on a knowledge test and calculated word frequencies in health consultation scenarios. They showed that the knowledge score was significantly higher in post-class than in pre-class. They concluded that there is a great need for nursing staff to administer physical and mental care to the public during a nuclear disaster.
General points
Although I agree the importance of education about radiation health sciences to nursing students, since they need to contribute to the risk communication with residents during a nuclear disaster such as the accident at TEPCO Fukushima Daiichi Nuclear Power Station, I have to point out that there is no novel scientific finding in this study. It is not surprising that students understood about radiation better after the lecture.
Also, authors should ask English correction to a native speaker.
Specific points
Line 85: The approval number of the ethics committee should be indicated.
Author Response
Response to Reviewer 1 Comments
Point 1:
Comments and Suggestions for Authors
Kawasaki et al. evaluated a basic training program to teach undergraduate nursing students how to deal with public anxiety following a nuclear disaster. They compared pre- and post-class response scores on a knowledge test and calculated word frequencies in health consultation scenarios. They showed that the knowledge score was significantly higher in post-class than in pre-class. They concluded that there is a great need for nursing staff to administer physical and mental care to the public during a nuclear disaster.
Response 1: Thank you for accepting the peer review.
Point 2:
General points:
Although I agree the importance of education about radiation health sciences to nursing students, since they need to contribute to the risk communication with residents during a nuclear disaster such as the accident at TEPCO Fukushima Daiichi Nuclear Power Station, I have to point out that there is no novel scientific finding in this study. It is not surprising that students understood about radiation better after the lecture.
Also, authors should ask English correction to a native speaker.
Response 2: Thank you for acknowledging the need for nursing students to study radiation health sciences.
As you pointed out, the fact that nursing students take lectures and learning is less novel. However, the novelty of this study is the learning method. Therefore, this study shows that the proposed learning method is effective. General nursing students have low readiness and motivation to learn radiation. Nursing students are highly interested in nursing care. "People's anxiety about radiation-related health" is used as a teaching material for consultation technology, which is nursing care. The novelty is that the introduction of radiation education will be implemented in a short period of time in line with the acquisition of skills to respond to consultations from residents. It is not a radiation nursing class. Certainly, the lessons proposed in this study are not a direct way to rescue people affected by radiation disasters. It is difficult for all nursing students in the basic education level to acquire highly specialized knowledge of radiation. This is a method of recognizing the need to learn about radiation in the basic nursing course leading to the next specialized learning. I also added it to the text.
Thank you for pointing out the need for having the English corrected by a native speaker. A native English speaker has already checked the submitted manuscript. However, when resubmitting, the author will have the language checked again by a native speaker.
Page 2, Lines 67-81
The Ministry of Education, Culture, Sports, Science, and Technology has indicated the need for a new curriculum in nursing education [24]. In the new curriculum, radiation nursing is included in disaster nursing practice. Nursing in the event of a disaster is not limited to radiation disasters. Not enough time compared to the content. Radiation-focused learning is not effective for pre-licensed nursing students because they did not consider basic physical learning to be directly related to nursing care. There is no clearly delineated method to teach nursing staff how to help increase this knowledge in the general public. As members of the public and also as future nursing staff, nursing students need to acquire fundamental knowledge about radiation and recognize pertinent disaster information and help the public in this regard [25]. After obtaining a nurse license, they need a foundation for additional learning about radiation expertise and nursing care.
The purpose of this study is to develop and evaluate methods for learning disaster-related radiation knowledge and care for nursing students attending Japanese universities. The focus of the method is to learn about radiation through nursing care, which is of great interest to nursing students.
Point 3
Specific points
Line 85: The approval number of the ethics committee should be indicated.
Response 3:
Thank you for pointing this out. The approval number has been added to "Materials and Methods" in the main text.
Page 3, Lines 98-99
“This study was conducted with the approval of the university’s ethics committee (Approval No. E-809-1).”
Reviewer 2 Report
In the abstract, the 111 3rd-year students need to be described as Nursing students.
The introduction needs delineation between the education of licensed/practicing nurses and the education of undergraduate/pre-licensure students.
The authors need to operationally define each of the learning domains. Psychomotor is described as "the ability to engage in activities to reduce the consultees' health anxiety." However the statements in the questionnaire that "I have thought about" are weak in this learning domain. Psychomotor is typically physical action, manipulating something, focusing on change and/or development in behavior or skills. "Thinking" about is not action-oriented.
Author Response
Response to Reviewer 2 Comments
Point 1:
Comments and Suggestions for Authors
In the abstract, the 111 3rd-year students need to be described as Nursing students.
Response 1:
Thank you for pointing out the need for a description of the subject in the abstract. It was clarified that the subjects were nursing students.
Page 1, Line 17
“We analyzed the data of 111 third-year nursing students attending a Japanese university.”
Point 2:
The introduction needs delineation between the education of licensed/practicing nurses and the education of undergraduate/pre-licensure students.
Response 2: A description of the Japanese curriculum has been added to the introduction.
Page 2, Lines 67-72
The Ministry of Education, Culture, Sports, Science, and Technology has indicated the need for a new curriculum in nursing education [24]. In the new curriculum, radiation nursing is included in disaster nursing practice. Nursing in the event of a disaster is not limited to radiation disasters. Not enough time compared to the content. Radiation-focused learning is not effective for pre-licensed nursing students because they did not consider basic physical learning to be directly related to nursing care.
Point 3:
The authors need to operationally define each of the learning domains. Psychomotor is described as "the ability to engage in activities to reduce the consultees' health anxiety.” However the statements in the questionnaire that "I have thought about" are weak in this learning domain. Psychomotor is typically physical action, manipulating something, focusing on change and/or development in behavior or skills. "Thinking" about is not action-oriented.
Response 3: Thank you for pointing this out.
As you mentioned, the psychomotor area is an area related to behavior and technology.
The subjects of this study were undergraduate nursing students and could not actually act during class. Therefore, the evaluation of the psychomotor area evaluated not the specific behavior or technique, but the recognition of the behavior or technique. I think that thinking about how to respond to consultations about radiation from residents among undergraduate students will lead to concrete actions in practice after starting to work in the field as a nurse. However, it is also important for undergraduate students to evaluate specific behaviors and skills, so I would like to recommend this for future research. Considering that it is a lesson evaluation, we revised the definition.
Page 3, Lines 110-111
Psychomotor Domain. Can think of actions to prepare for a nuclear disaster; think of ways to address people's health concerns.
Reviewer 3 Report
Manuscript ID: ijerph-871179
Document: Addressing radiation-induced health concern: Evaluating a nursing course on nuclear disaster management
This study conducted a questionnaire survey to undergraduate nursing students to examine the effect of a special-purpose class on change of three domains of educational goals. While the used appropriate logical methods for investigating the effects. it remains unclear whether the study aims to clarify the effect of the class for educating radiation knowledge, or to seek how to consult concerned people. The reason would be that the methods of study used three domain of educational goals. The domains would be intercorrelated each other and not easy to simply clarify the effects. However, this study dealt with only 90 min class for undergraduate students. The increase of knowledge points after the class may be very natural and does not seem to provide a meaningful result. Interesting results would be comparison analysis of the words between before and after watching the video. On these bases, it will be required to improve the document to reflect the original aims and also to more analyze the results of the study.
General comments:
- The title does not appropriately express the content, and thus seems to be misleading to readers. The following title would be recommended:
Analysis of the training effect of nursing undergraduate course for radiation-related health concerns – a single group experiment
- The study aims to evaluate a basic training program to achieve educational goals that consist of three domains such as understanding of radiation, imagination of concerned people and motivation of a nuclear disaster. However, for testing understanding, the study used knowledge test that includes a 27-items comprising 20 true-false items. All of the 27-items should be described in Appendix, because three items of “Cognitive” in Table 1 are unclear. If the level of understanding of radiation is investigated, learning of radiation physics, biology and health effects/risk including radiation regulation may be fundamental, but the authors may be aware that the training program only for 90 min cannot achieve the goal of the first learning domain. If so, the authors should discuss what to teach to undergraduate students in order to consult people with radiation-related health concerns. In 4.1, the authors should discuss more specific knowledge for the example scenario on internal exposure, at least in order to enable to conduct consultation.
- Similarly above, 12 items were questioned for two domain of “Affective “ and “Psychomotor”. All of the 12 items should be described in Appendix, because three items of “Cognitive” in Table 1 are unclear.
- Figure 2 and 3 shows the result of the words students answered in a free-text way. The comparison between two looks similar. The result would be very surprising to me, because social-economical-ethical-related words were little found. The authors addressed metabolism and emission newly appeared after the class. The scenario of internal exposure would be associated with emergence of the words. However, the readers could not understand what happened after the class. The authors should provide more detailed contents of the lecture talking about the basic knowledge about radiation, which would affect the words responded by students.
- What I want to know is specificity of the nursing student of Hiroshima University. They may know very well the victims of the atomic bomb and have experienced negative image, sometime biased information. These background behind the class would not be negligible to this study. The authors need to discuss the influence of the atomic bomb history in Hiroshima even if you did not consider these issues in this study.
Specific comments
- Statistical analysis
The study conducted a paired t-test for comparison of knowledge test. It is not appropriate to use a paired t-test of the biased distribution of the score.
- Line80; “ undergraduates are three months into the specialized course on nursing care” are unclear.
- Line 62-69; discussed some issues of radiation education for undergraduates. Furthermore, the authors may fail to consider what is radiation education or disaster education. Radiation and radioactivity are widely used in medicine. Recently, radiation education in nursing education has been emphasized after the Fukushima, but medical exposure should be more focused in terms of radiation education. The authors should describe current situations of radiation education on new curriculum recommended by MEXT.
Author Response
Response to Reviewer 3 Comments
Point 1:
Comments and Suggestions for Authors
Manuscript ID: ijerph-871179
Document: Addressing radiation-induced health concern: Evaluating a nursing course on nuclear disaster management
This study conducted a questionnaire survey to undergraduate nursing students to examine the effect of a special-purpose class on change of three domains of educational goals. While the used appropriate logical methods for investigating the effects. it remains unclear whether the study aims to clarify the effect of the class for educating radiation knowledge, or to seek how to consult concerned people. The reason would be that the methods of study used three domains of educational goals. The domains would be intercorrelated each other and not easy to simply clarify the effects. However, this study dealt with only 90 min class for undergraduate students. The increase of knowledge points after the class may be very natural and does not seem to provide a meaningful result. Interesting results would be comparison analysis of the words between before and after watching the video. On these bases, it will be required to improve the document to reflect the original aims and also to more analyze the results of the study.
Response 1: Thank you for your specific suggestions regarding the ambiguity of the research purpose.
It consists of learning radiation knowledge for nursing care (consultation). Therefore, the goals to be achieved are set to relate to both sides. General nursing students have low readiness and motivation to learn radiation. However, nursing students are very interested in nursing care.
Using "People's anxiety about radiation health" as a teaching material for consultation technology, it was intended for students to acquire the minimum knowledge and skills to respond to consultations on radiation from residents in a short time.
When it comes to the knowledge necessary for nursing care, I thought that the learning effect of esoteric radiation knowledge would increase.
As you pointed out, the number of words related to radiation has increased. The introduction and considerations have been revised to reflect these.
Page 2, Lines 78-81
The purpose of this study is to develop and evaluate methods for learning disaster-related radiation knowledge and care for nursing students attending Japanese universities. The focus of the method is to learn about radiation through nursing care, which is of great interest to nursing students.
Page 9, Lines 271-280
The number of people using nouns related to radiation increased and the nouns metabolism and emission newly appeared. It indicates that they have gained knowledge of the metabolism of radioactive substances in internal exposure, that is, the biological half-life. Nouns like worry and fuan (unease/worry/anxiety) represent an emotional dimension. The dramatic increase in the number of students using such words after the class suggests that more students sympathized with the consultee in the example scenario. Students were deemed to have achieved the attainment goals upon the qualitative evaluation of their response to the consultee [40]. In the Hiroshima atomic bomb, the initial radiation indicated by "Pika" has been regarded as a health risk, but the risk of internal exposure due to sand dust has also been taken up [41]. Cases of internal exposure are considered important in future nuclear disasters.
Point 2:
General comments:
The title does not appropriately express the content, and thus seems to be misleading to readers. The following title would be recommended:
Analysis of the training effect of nursing undergraduate course for radiation-related health concerns – a single group experiment
Response 2:
Thank you for pointing this out. We have corrected it according to the title you proposed.
Page 1, Line 2
“Analysis of the training effect of nursing undergraduate course for the management of radiation-related health concerns – a single group experiment”
Point 3:
The study aims to evaluate a basic training program to achieve educational goals that consist of three domains such as understanding of radiation, imagination of concerned people and motivation of a nuclear disaster. However, for testing understanding, the study used knowledge test that includes a 27-items comprising 20 true-false items. All of the 27-items should be described in Appendix, because three items of “Cognitive” in Table 1 are unclear. If the level of understanding of radiation is investigated, learning of radiation physics, biology and health effects/risk including radiation regulation may be fundamental, but the authors may be aware that the training program only for 90 min cannot achieve the goal of the first learning domain. If so, the authors should discuss what to teach to undergraduate students in order to consult people with radiation-related health concerns. In 4.1, the authors should discuss more specific knowledge for the example scenario on internal exposure, at least in order to enable to conduct consultation.
Similarly above, 12 items were questioned for two domain of “Affective “ and “Psychomotor”. All of the 12 items should be described in Appendix, because three items of “Cognitive” in Table 1 are unclear.
Response 3: The cognitive domain is not just knowledge, so it has three goals. In 4.1, we added the examination of the minimum knowledge to answer the consultation of the case. All goals are listed in Table 1. As you pointed out, knowledge of radiation cannot be learned in 90 minutes.
The Radiation Nursing Society has published 15 lessons. However, many nursing colleges cannot afford to treat radiation nursing as a subject, independently. There is increasing learning about priority diseases and treatments. The focus of this study is on recognizing the need for students as a basis for advancing to the next stage of specialized radiation learning.
We have extracted the minimum knowledge as a consultation in areas where it is not necessary to consider the effects of the nuclear accident. After that, it will be the basis for students to research it on their own and when they are dispatched. We touch on this goal in the cognitive domain. Their ability is not sufficient to step into internal exposure. Tests have been added as additional material.
Page 8, Lines 202-209
In this domain, the focus of the assessment was whether the student had acquired the basic radiation knowledge needed to reduce consultees’ health anxiety. We had selected the knowledge on radiation for students to reduce their health concerns as follows. Radioactive substances (iodine-131 and cesium-137) and radiation, unit (㏃, ㏜), internal exposure, half-life, metabolism, hot spot, dosimetry, means and place for collecting information. We added stable iodine tablet, as a precautionary measure. For student motivation, it is important to strictly select the knowledge necessary for case consultation. Choosing the contents needed for a case from a lot of knowledge is the next level.
Point 4:
Figure 2 and 3 shows the result of the words students answered in a free-text way. The comparison between two looks similar. The result would be very surprising to me, because social-economical-ethical-related words were little found. The authors addressed metabolism and emission newly appeared after the class. The scenario of internal exposure would be associated with emergence of the words. However, the readers could not understand what happened after the class. The authors should provide more detailed contents of the lecture talking about the basic knowledge about radiation, which would affect the words responded by students.
Response 4: As you pointed out, the language used by students changed depending on the lecture.
The case was regarding children inhaling dust when playing sports at school. Three months had passed since the accident, and the distance is 100 km away. Current standards indicate that there are no health effects. For the content of the lecture, we chose the unit of radiation, the type of exposure, and so on. Therefore, we could relate it to the case.
To resolve parental concerns, students needed to explain health impact criteria and children's metabolism, and the number of relevant words increased. Our goal had been achieved.
The lecture content has been added to the method for the reader to understand.
Internal exposure to dust is seen as an issue. By current standards, there is little risk to the consultant, but it is a concern after the nuclear accident. If we chose one case from many situations, we chose internal exposure to dust. Thinking about answers to reduce the anxiety of the counselor gives the student some knowledge.
Page 4, Lines 125-129
2.3.4 Lecture
The contents of the lecture are shown below.
Radioactive substances (iodine-131 and cesium-137) and radiation, unit (㏃, ㏜), natural radiation and artificial radiation, internal exposure and external exposure, half-life, metabolism, stable iodine tablet, hot spot, dosimetry, means and place for collecting information.
Page 9, Line 264
The number of people using nouns related to radiation increased and the nouns metabolism and emission newly appeared. It indicates that we have gained knowledge of the metabolism of radioactive substances in internal exposure, that is, the biological half-life.
Point 5:
What I want to know is specificity of the nursing student of Hiroshima University. They may know very well the victims of the atomic bomb and have experienced negative image, sometime biased information. These background behind the class would not be negligible to this study. The authors need to discuss the influence of the atomic bomb history in Hiroshima even if you did not consider these issues in this study.
Response 5: Thank you for your question regarding the background of the research subjects. As you pointed out, the fact that the research subject is a student at Hiroshima University may have influenced the research results. However, many students at Hiroshima University are not only from Hiroshima prefecture, but also from outside the prefecture. It cannot be said that there are many students who are particularly interested in radiation.
Furthermore, it seems that some students have learned about the atomic bombing of Hiroshima after entering university, but I do not have the impression that many students are thinking about the atomic bombing of Hiroshima and radiation nursing. Regarding this, we believe that the research subjects are students of Hiroshima University will not have a significant impact on the research results.
I also added the situation in Hiroshima.
Page 9, Lines 278-280
In the Hiroshima atomic bomb, the initial radiation indicated by "Pika" has been regarded as a health risk, but the risk of internal exposure due to sand dust has also been taken up [41]. Cases of internal exposure are considered important in future nuclear disasters.
Point 6:
Comments:
Specific comments
Statistical analysis
The study conducted a paired t-test for comparison of knowledge test. It is not appropriate to use a paired t-test of the biased distribution of the score.
Response 6: Thank you for pointing out the analysis method. The analysis method was changed using the Wilcoxon signed rank test for pre- and post-knowledge comparison. The Wilcoxon signed rank test gave the same results as the t-test. The result of the text has been corrected as follows.
Page 5, Line 161
The knowledge and self-assessment tests were analyzed with SPSS version 25, using Wilcoxon signed rank test, with the level of significance set at 0.05. For the analysis of the worksheets, we used IBM SPSS Text Analytics for Surveys 4.0.
Page 5 Lines 165-166
“The combined pre-class mean score was 12.2 points and the post-class mean score was 21.0, demonstrating a significant increase using Wilcoxon signed rank test (p < 0.001)”
Point 7:
Line80; “ undergraduates are three months into the specialized course on nursing care” are unclear.
Response 7:Thank you for your question regarding the research subjects. In Japanese universities, first- and second-year students study liberal arts subjects and an introduction to nursing science. They learn about specialized nursing care related to clinical nursing from the third year. Line 80 means it has been three months since I started learning specialized nursing care related to clinical nursing.
Page 2, Lines 93 and Page 3, Line 93
During the third year, the month of July was three months since nursing undergraduates started studying specialized nursing care related to clinical nursing.
Point 8:
Line 62-69; discussed some issues of radiation education for undergraduates. Furthermore, the authors may fail to consider what is radiation education or disaster education. Radiation and radioactivity are widely used in medicine. Recently, radiation education in nursing education has been emphasized after the Fukushima, but medical exposure should be more focused in terms of radiation education. The authors should describe current situations of radiation education on new curriculum recommended by MEXT.
Response 8: As you pointed out, medical exposure is an important component of radiation education; however, time constraints make it difficult to implement curriculum-based undergraduate education that includes specialized radiation nursing as an independent subject. For this reason, the Ministry of Education, Culture, Sports, Science and Technology and the Japanese Society of Radiation Nursing has recommended that it be included as an independent subject in undergraduate education, but this has not yet been actualized. Studying the independent subject of radiation nursing is available to radiation specialist nurses after 5 years of work experience.
In Japan, the Ministry of Education, Culture, Sports, Science and Technology has included it in disaster education as a response in the event of a radiation disaster. An explanation about the curriculum of disaster education and radiation education was added.
Page 2, Lines 67-72
The Ministry of Education, Culture, Sports, Science, and Technology has indicated the need for a new curriculum in nursing education [24]. In the new curriculum, radiation nursing is included in disaster nursing practice. Nursing in the event of a disaster is not limited to radiation disasters. Not enough time compared to the content. Radiation-focused learning is not effective for pre-licensed nursing students because they did not consider basic physical learning to be directly related to nursing care.
Round 2
Reviewer 1 Report
Authors sufficiently revised a manuscript, according to the suggestion of a reviewer. I think that now this manuscript should be accepted.
Author Response
Thank you for your careful peer review.
Reviewer 3 Report
The revised document has been improved according to the comments. However, further revision is needed according to the following specific comments.
Specific comments:
Line 121-125: The following is better to be described:
The key contents of the lecture are radioactive substances.........,and how to collect the relevant information.
Line 198-201: The following is better to be described:
The selected key contents are radioactive substances.........,and how to collect the relevant information.
Line 200-201: We added .... measures. ->
Iodine thyroid blocking using stable iodine tablet was added to the key contents.
Line 265-266: Improve like: This indicates that they gained some knowledge of the behavior of radioactive substances following internal exposure by understanding the biological half-life.
Line 270-273: This added sentence looks scientifically inappropriate. This sentence can be interpreted that internal exposure are a key in the atomic bomb health consequence in Hiroshima. The news information of Ref.(41) is inappropriate reference and should be removed, because this information would lead to misunderstanding of current scientific consensus. The first comment from the reviewer was that students in Hiroshima have been influenced by a lot of information and voices from people regarding the atomic bomb disaster. Radiation education should consider the consequence from risk perception. In your study, these things remain to be clear. However, I agree that undergraduate education should focus on fundamental concepts and knowledge. This paper provides how to educate the radiation basics.
Author Response
Comments and Suggestions for Authors
The revised document has been improved according to the comments. However, further revision is needed according to the following specific comments.
Response: Thank you for your careful peer review.
Specific comments:
1)Line 121-125: The following is better to be described:
The key contents of the lecture are radioactive substances........., and how to collect the relevant information.
Response: Thank you for suggesting the appropriate English expressions. We have made the revisions. Please refer to Line numbers 122-126.
2)Line 198-201: The following is better to be described:
The selected key contents are radioactive substances........., and how to collect the relevant information.
Response: We have fixed it on your advice. Please refer to line numbers 198-201.
3)Line 200-201: We added .... measures. ->
Iodine thyroid blocking using stable iodine tablet was added to the key contents.
Response: We have fixed it on your advice. Please refer to line numbers 201.
4)Line 265-266: Improve like: This indicates that they gained some knowledge of the behavior of radioactive substances following internal exposure by understanding the biological half-life.
Response: We have fixed it on your advice. Please refer to Line numbers 266-268.
5)Line 270-273: This added sentence looks scientifically inappropriate. This sentence can be interpreted that internal exposure are a key in the atomic bomb health consequence in Hiroshima. The news information of Ref.(41) is inappropriate reference and should be removed, because this information would lead to misunderstanding of current scientific consensus. The first comment from the reviewer was that students in Hiroshima have been influenced by a lot of information and voices from people regarding the atomic bomb disaster. Radiation education should consider the consequence from risk perception. In your study, these things remain to be clear. However, I agree that undergraduate education should focus on fundamental concepts and knowledge. This paper provides how to educate the radiation basics.
Response: As you pointed out, this is a new hypothesis which states that internal exposure can explain parts that cannot be explained by previous findings.
This paper is a proposal that nursing students can work on radiation learning without any resistance. As it is out of the scope of this dissertation, we have deleted it.